# A multi-dimensional incomplete stepped-wedge trial design to estimate the impact of standards-based audit

**Sarah Ann White** [ORCID]**\*, Florence Mgawadere**[¤]

Department of International Public Health, Liverpool School of Tropical Medicine, Liverpool, United Kingdom

¤ Current address: Institute of Population Health, School of Health Sciences, University of Liverpool, Liverpool, United Kingdom
* sarah.white@lstmed.ac.uk

**Data Availability Statement:** All relevant data are within the paper.

**Funding:** NvdB received funding for this study from Unité de Gestion de Projets with financing from The Global Fund to Fight AIDS, Tuberculosis

## Abstract

A clinical audit is a low-cost process used for quality improvement in healthcare. Such audits are however infrequently used in resource poor countries, where the need for and potential impact of quality improvement is higher. Sets of standards for use in maternal and newborn care have been established based on internal guidelines and evidence. The before-after design of a clinical audit is prone to bias in the estimation of the impact of conducting a clinical audit. A trial design that would provide an unbiased estimate of the impact of implementing a clinical audit process on the attainment of standards selected (a standards-based audit) was needed. The aim of this paper is to introduce and describe the design of trials we developed to meet this need. A novel randomised stepped-wedge trial design to estimate the impact of conducting standards-based audits is presented. A multi-dimensional incomplete stepped-wedge cluster randomised trial design suitable for estimation of the impact of Standards-based audits on compliance with standard is proposed; two variants are described in detail. A method for sample size estimation is described. Analyses can be performed for the binary outcome using a generalised linear mixed model framework to estimate the impact of the approach on compliance with standards subjected to a standards-based audit; additional terms to consider including in sensitivity analyses are considered. The design presented has the potential to estimate the impact of introducing the standards-based audit process on compliance with standard, while providing participating healthcare providers opportunity to gain experience of implementing the standards-based audit process. The design may be applicable in other areas in which multiple processes are to be studied.

## Introduction

### Clinical audits

A clinical audit is a low-cost quality-improvement process which often has the potential to improve the quality of clinical care. This quality-improvement process seeks to improve

and Malaria under Grant Agreements TGO- H-PMT No. 1467, TGO-T-PMT No. 1468 and TGO-M-PMT No. 1469. The funders had no role in study design, data collection and analysis, decision to publish or preparation of the manuscript.

**Competing interests:** The authors have declared that no competing interests exist.

patient care and outcomes by systematically reviewing care against explicit standards, with identification and implementation of changes needed to achieve the desired standard of care [1]. The World Health Organisation [1] includes the descriptor "criterion-based" to distinguish from critical incident audits such as maternal death reviews. Clinical audits are usually performed in clinical practice in a single healthcare facility by healthcare workers themselves. Graham and colleagues [2] outline five steps in a clinical audit cycle of hospital based obstetric care in the management of five obstetric complications (Table 1). A data collection tool is used to capture the data required to determine if the care provided to each sampled client attained the standard or not. Review of the detailed data facilitates identification of potentially useful changes to implement in step 4. The percentage of clients whose care attained the desired standard during step 2 is the reference point to which the level attained in step 5 is compared. In practice, a clinical audit may be iterative, with additional cycles reviewing the quality involving additional remedial actions to address persistent deficiencies in the quality attained [3]. A clinical audit is inherently an uncontrolled before-after design. Since such designs are not able to quantify the impact of any contemporaneous independent changes on the outcome they are prone to bias in the derivation of effect estimates. The persistently poor maternal outcomes in low- and middle-income countries indicate a need for quality improvement. Although clinical audit has been used in a number of low- and middle-income countries [4,5] its deployment remains infrequent.

## Standards for maternal and newborn care

A set of 25 standards of care has been developed and agreed, for clinical content specific to Emergency Obstetric and Newborn Care, and for respectful care [1]. The standards were developed with inputs from a diverse group of international stakeholders and contributors. The standards cover the whole spectrum of care that should be provided to women and their newborns and can be used to audit clinical practice in a systematic and objective way. For each standard the care received by clients can be assessed to determine if the defined standard was attained or not for each client. Standards vary in complexity, from requiring that a single piece of information is checked, to multiple fields of information. For example, "*As part of active management of the third stage of labour, all women who give birth at the healthcare facility receive oxytocin after birth*" requires only that case notes be checked to determine if receipt of oxytocin was documented. By contrast "*Every woman in labour in a healthcare facility is monitored using a partograph correctly*" requires that the existence of a partograph be determined and that multiple details within it be checked. For each standard the outcome of interest was whether the care provided to each individual client to which the standard applies was

**Table 1. Steps of a clinical audit cycle and a standards-based audit.**

| Steps of a clinical audit cycle | Activity[a] | Steps of a Standards-based audit | Status in M-DISW-CRT |
|---|---|---|---|
| 1 | **Establish criteria for best practice in detection and management of defined obstetric complications** | 0 | |
| 2 | **Observe current practice** and estimate the proportion of clients in which the standard was attained | 1 | Control |
| 3 | **Feedback findings** | 2 | Transition |
| 4 | **Implement changes in practice** to address deficiencies | | |
| 5 | **Re-evaluate practice and feedback** | 3 | Intervention |

a Descriptions in bold font are extracted from Fig 2 of reference **2.**

compliant with the standard or not. We adopted the terminology Standards-based audit (S-BAs) for the use of these defined standards in the clinical audit process. Table 1 indicates the taxonomy we use in describing S-BA in relation to that of clinical audits.

## Trial design aims

Our work with Ministries of Health in several countries in Africa provided opportunity to design a multi-centre trial, in multiple facilities providing antenatal, postnatal and newborn care services, that would address the limitations of the before-after study design. Within several countries we were able to design and conduct a trial within the public health care system.

We wanted a randomised controlled trial design to estimate the benefit to facilities of adopting the practice of clinical audit of the standards developed, in order to provide an evidence-base for advocating for the adoption of this approach.

Prior to the trial, the level of compliance with each standard, was expected to vary between standards, and not to be consistent across facilities. A major secondary aim of our work was to equip the Health Care Providers (HCPs) within participating facilities with the knowledge and competence to adopt the practice of performing S-BA in their clinical practice beyond the duration of the trial. We therefore wanted a design which both incorporated audit cycles within each facility and allowed the HCPs to select the standards to be audited within their facility that they considered most in need of improving. Adoption of the process of using S-BA was the focus, rather than any particular standard. When used in clinical practice in a single facility a S-BA is inherently a before-after study, with the associated limitations arising for underlying secular changes to potentially impact change in compliance.

A clinical audit inherently involves repeated cross-sectional collection of data, and a change of status, from control (standard of care in step 1 of S-BA) to intervention (re-evaluation of practice in step 3 of S-BA) status. The inherent transition from control to intervention status suggested that an incomplete stepped-wedge cluster randomised trial design (SW-CRT) may be a suitable one to consider in seeking to evaluate the impact of conducting S-BAs in multiple facilities (clusters) [6–9]. SW-CRT designs provide a structure which accounts for any underlying changes over time when estimating the intervention effect, which before and after studies are not able to do. In the proposed design data collection in each participating facility is expanded, for each standard audited, beyond that of a standard clinical audit. The inclusion of multiple facilities enables randomisation to be performed, thereby providing opportunity for the derivation of estimates that account for any underlying secular trend. The status of each of the steps of a S-BA in the proposed design is also indicated in Table 1. If a single standard was the focus a staircase design [10] might be used.

During the last three decades, stepped-wedge trial designs have increasingly been used in the evaluation of interventions within health systems. The study period is divided into a sequence of pre-defined time periods, the points of transition between them are referred to as steps. An inherent characteristic of trials using this design is that for each participating cluster there is a specified step, at which the cluster ceases to be in the control state and moves either instantaneously or after a transition period, into the intervention state [11]. Once a cluster has entered the intervention state the control state cannot be resumed. This makes the design an attractive one for use in health systems research when an intervention is expected to be of benefit, there are logistic constraints on the roll out of the intervention and an estimate of the impact of the intervention is desired [6,8].

## Modifications of Stepped Wedge trial designs in literature

Hemming, Lilford and Girling [12] outline several modifications of a SW-CRT, including an incomplete cross-sectional design. Lyons, Li, Hughes and Rowhani-Rahbar [13] identified

**Table 2. Status in each step of a complete factorial stepped wedge design for 2 interventions, with 4 steps, by sequence.**

| Sequence (Group of clusters) | Step | | | | |
|---|---|---|---|---|---|
| | **0** | **1** | **2** | **3** | **4** |
| 1 | *(0,0)* | *(1,0)* | *(1,0)* | *(1,1)* | *(1,1)* |
| 2 | *(0,0)* | *(0,0)* | *(1,0)* | *(1,0)* | *(1,1)* |
| 3 | *(0,0)* | *(0,0)* | *(0,1)* | *(0,1)* | *(1,1)* |
| 4 | *(0,0)* | *(0,1)* | *(0,1)* | *(1,1)* | *(1,1)* |

(a,b) indicates the state for interventions A and B respectively in the step, 0 = control, 1 = intervention.

Based on Fig 2 part D of reference **14**.

For consistency with literature on the SW-CRT design 0 is used for control periods and 1 for intervention periods, whereas in clinical audit literature it is more conventional to use 1 and 2 respectively to code these assessments.

four variants of the SW-CRT that allow two interventions to be assessed within a single cluster randomized trial. The design we present is a multi-dimensional incomplete cross-sectional design, in contrast to the factorial SW-CRT of Lyons and colleagues [13]. In their factorial SW-CRT each cluster is assigned to adopt both interventions, with the status for each individual intervention following a stepped wedge design, the sequence in which the interventions are introduced randomised and a single outcome measured. Table 2 indicates the status over time for each group of clusters in the design of Lyons and colleagues [13]. To motivate the proposed design let (a,b) denote the control / intervention status at a facility with regard to interventions A and B, where 0 indicates control state and 1 indicates intervention state. Thus, in sequence 1 intervention A starts in step 1, and intervention B in step 3. In the factorial design data are collected for each cluster in each time-period of the trial for the single outcome which is thought to be impacted by each of the interventions. Although multiple interventions are involved the single outcome makes this a uni-dimensional design.

The aim of this paper is to introduce and describe the design of trials we developed to evaluate the impact of introducing the process of conducting Standards-based audits (S-BA) on the quality of maternal and newborn care in developing countries.

## Multi-dimensional incomplete stepped-wedge trial designs

The design we present has the following components:

i. Multiple health facilities participate and each facility conducts a sequence of S-BAs with an allocated schedule. Data may additionally be collected for each standard in some other study steps.

ii. Health facilities are randomised to the time period (step or month) in which they begin S-BA cycles.

iii. Each S-BA cycle comprises three consecutive time periods (1: assessment of current compliance with standard; 2: review findings and implement changes to address deficiencies; 3: re-assess compliance with standard).

iv. Each health facility identifies the standards of highest priority for them to audit, based on the perceived level of compliance with each standard.

v. The sequence in which each facility audits standards is randomised and balance is optimised overall across months for each standard.

vi. For each standard a common tool is used across all facilities auditing that standard, to assess compliance of the care provided with the standard, for 25 clients in each step which is assessed.

Although this design has some similarities to the uni-dimensional factorial SW-CRT it is multi-dimensional. In a factorial SW-CRT the same outcome(s) are assessed for each intervention; in the proposed design each standard introduces another dimension to the data collected. In the factorial design all possible combinations of all interventions occur, with each cluster expected to move from the absence of all interventions at the start of the trial to the presence of all interventions at the end of the trial. This is not required of the proposed multi-dimensional incomplete SW-CRT design (M-DISW-CRT), although it is true of some designs.

## Design variants

The proposed design has several possible variants. We describe two variants in detail. In design A, audit cycles are completed consecutively in three-month phases, without overlap. In the alternative design, B, audit cycles overlap, with month 3 of the audit cycle for the first standard being month 1 of the audit cycle for the next standard audited. To illustrate these designs, we suppose that there are six standards, identified as $S_a$, $S_b$, $S_c$, $S_d$, $S_e$ and $S_f$.

### Design A

For this design, suppose that standards, $S_a$, $S_b$, $S_c$ and $S_d$, are of interest in the trial and that each facility that participates in the trial is expected to complete two S-BAs, in two consecutive phases, e.g. in months 1–3 and months 4–6. Within each S-BA data are collected from a sample of 25 clients and reviewed in the first and third months. In order for an M-DISW-CRT to be applied to S-BAs when the cycles do not overlap it is necessary that facilities transition from control to intervention state in a staggered manner to avoid fully confounding step with intervention. Therefore, in design A the participating facilities are randomised to three strata which determine the month in which S-BAs are to start: month 1, 2 or 3. Data collection for the trial at each facility is completed within six calendar months and the total trial duration is eight months.

Randomisation is used to determine the sequence in which the standards selected by each facility are audited. Suppose that there are 40 facilities, of which 28, 21, 17 and 14 select standard $S_a$, $S_b$, $S_c$ and $S_d$ respectively, with the distribution across the six possible pairings selected as indicated in Table 3A. A focus in randomisation is to balance the distribution of the standards between the two phases; Table 3B shows one of the two possible allocation choices to optimise this balance. For each standard there would be 6 possible timings for conducting a S-BA for the standard, determined by the three strata (months) for commencing audit cycles and whether the standard is audited in the first or the second audit cycle. Ideally, to minimise the bias arising from any order effects in estimation, there would be balance for each standard

**Table 3. a Frequency distribution of pairs of standards selected by facilities.** b Cross-tabulation of an allocation of facilities to sequences of standards.

| Pair of standards | Number selecting | First phase | Second phase | | | | Total |
|---|---|---|---|---|---|---|---|
| | | | $S_a$ | $S_b$ | $S_c$ | $S_d$ | |
| $S_a$ and $S_b$ | 14 | $S_a$ | | 7 | 4 | 3 | 14 |
| $S_a$ and $S_c$ | 8 | $S_b$ | 7 | | 2 | 2 | 11 |
| $S_a$ and $S_d$ | 6 | $S_c$ | 4 | 2 | | 2 | 8 |
| $S_b$ and $S_c$ | 4 | $S_d$ | 3 | 1 | 3 | | 7 |
| $S_b$ and $S_d$ | 3 | **Total** | 14 | 10 | 9 | 7 | 40 |
| $S_c$ and $S_d$ | 5 | | | | | | |

| Sequence | Stratum | Phase audited | Month | | | | | | | |
|---|---|---|---|---|---|---|---|---|---|---|
| | | | 1 | 2 | 3 | 4 | 5 | 6 | 7 | 8 |
| 1 | I | 1 | 0 | T | 1 | | | 1 | | |
| 2 | II | 1 | | 0 | T | 1 | | | 1 | |
| 3 | III | 1 | | | 0 | T | 1 | | | 1 |
| 4 | I | 2 | 0 | | | 0 | T | 1 | | |
| 5 | II | 2 | | 0 | | | 0 | T | 1 | |
| 6 | III | 2 | | | 0 | | | 0 | T | 1 |

**Fig 1. Schematic representation for Design A of data collection and action for a single standard.** 0 (in pale yellow cells) denotes assessment of compliance with the standard under the current standard of care. T (in pale grey cells) denotes a transition month in which action is taken after review of data collected previously. 1 (in orange cells) denotes assessment of compliance after taking action to improve the quality of care delivery for the standard. Within each sequence the audit cycle phases are indicated by boldly bordered sets of three-month periods.

across the six groups thus defined. Complete balance can only be achieved by some reallocation of standards to be audited so that the number auditing each standard is a multiple of 6. However, since this risks assigning a facility to audit a standard which already has good compliance it is not recommended.

In the first month of data collection at a facility additional data are collected; as well as data for the standard being audited in the first phase, data for the standard to be audited in the second phase are also collected. Similarly, in the last month of data collection at a facility data are collected for the standard audited in the first phase as well as the one audited in the second phase. Fig 1 shows the timings of data collection for a single standard for each of the six possible sequences. All data are included in the analysis (as discussed under "Inclusion of baseline and endline assessments")

To optimise the engagement of HCPs with the trial the pair of standards audited at any facility depends on the priorities of the facility. Some imbalance in the frequency with which each standard is audited is therefore expected. For the example provided in Table 3A 28 facilities chose to audit $S_a$; of these fourteen also chose to audit $S_b$, eight chose $S_c$, and six chose $S_d$.

Extending and modifying the notation used for Table 2, let (a,b,c,d) denote the control / intervention status at a facility with regard to $S_a$, $S_b$, $S_c$ and $S_d$, where 0 again indicates control state, t indicates transition state and 1 indicates intervention state. Italicised emboldening indicates that data are collected for that standard in that month. Table 4 shows the status and months of data collection for facilities which audit $S_a$ and $S_b$ (Sequences 1 to 6) and the last two sequences (35 and 36) that audit $S_c$ and $S_d$. For any given pair of standards there are six sequences to which the facilities are randomised. For pairs selected by six or more facilities at least one facility is allocated to each sequence for that pair, whereas for pairs selected by fewer

**Table 4. Status in each Step of a four-dimensional incomplete stepped wedge design with 6 steps, by sequence for pairs of standards.**

| Pair of standards | Sequence | Step | | | | | | | |
|---|---|---|---|---|---|---|---|---|---|
| | | 1 | 2 | 3 | 4 | 5 | 6 | 7 | 8 |
| S_a and S_b | 1 | (***0***,***0***,0,0) | (t,0,0,0) | (***1***,0,0,0) | (1,***0***,0,0) | (1,t,0,0) | (***1***,***1***,0,0) | | |
| | 2 | | (***0***,***0***,0,0) | (t,0,0,0) | (***1***,0,0,0) | (1,***0***,0,0) | (1,t,0,0) | (***1***,***1***,0,0) | |
| | 3 | | | (***0***,***0***,0,0) | (t,0,0,0) | (***1***,0,0,0) | (1,***0***,0,0) | (1,t,0,0) | (***1***,***1***,0,0) |
| | 4 | (***0***,***0***,0,0) | (0,t,0,0) | (0,***1***,0,0) | (***0***,2,0,0) | (t,1,0,0) | (***1***,***1***,0,0) | | |
| | 5 | | (***0***,***0***,0,0) | (0,t,0,0) | (0,***1***,0,0) | (***0***,1,0,0) | (t,1,0,0) | (***1***,***1***,0,0) | |
| | 6 | | | (***0***,***0***,0,0) | (0,t,0,0) | (0,***1***,0,0) | (***0***,1,0,0) | (t,1,0,0) | (***1***,***1***,0,0) |
| | : | : | : | : | : | : | : | : | |
| S_c and S_d | : | : | : | : | : | : | : | : | |
| | 35 | | (0,0,***0***,***0***) | (0,0,0,t) | (0,0,0,***1***) | (0,0,***0***,1) | (0,0,t,1) | (0,0,***1***,***1***) | |
| | 36 | | | (0,0,***0***,***0***) | (0,0,0,t) | (0,0,0,***1***) | (0,0,***0***,1) | (0,0,t,1) | (0,0,***1***,***1***) |

(a,b,c,d) indicates the state of S_a, S_b, S_c and S_d in the step, 0 = control, t = transition, 1 = intervention.

Italicised emboldening indicates data collection to assess compliance for the standard.

All sequences are shown for facilities auditing S_a and S_b and the last two sequence for facilities auditing S_c and S_d.

than six facilities some sequences are not assigned to any facilities. Fig 2 indicates the data to be collected for a facility in sequence 1. For the four other pairs of standards the schematic representation is similar. Since this trial design involves four standards this is a four-dimensional incomplete SW-CRT.

## Design B

For this design we suppose that 18 facilities are to participate, all six standards are of interest and each facility that participates in the trial is expected to be able to complete six S-BAs, one for each standard, in six overlapping phases, e.g. in months 1–3, 3–5, 5–7, 7–9, 9–11 and 11–13. Details of an audit cycle, stratification and randomisation for this design are as in Design A, except that there are only two levels of stratification for month of starting audit cycles (1 or 2) and all facilities will audit all standards. To ensure balance over time and avoid the risk of order effects randomisation is used to ensure that each standard is audited as equally as

| Representation | Standard | Month (Phase 1) | | | Month (Phase 2) | | |
|---|---|---|---|---|---|---|---|
| | | 1 | 2 | 3 | 4 | 5 | 6 |
| By Standard | S_a | X(a) | T(a) | X(a) | | | X(a) |
| | S_b | X(b) | | | X(b) | T(b) | X(b) |
| As a facility | S_a and S_b | X(a,b) | T(a) | X(a) | X(b) | T(b) | X(a,b) |

**Fig 2. Schedule of assessments for a facility in sequence 1 in Design A.** X(a) indicates that data for S_a was to be collected in that month. T(a) indicates that data for S_a was to be reviewed and action taken. X(a,b) indicates that data for S_a and S_b was to be collected in that month. X(b) and T(b) are similarly defined. Pale yellow and orange cells respectively indicate measurements obtained under the current standard of care and after action. Pale grey cells indicate months in which action is taken.

| Row | Square I | | | | | | Square II | | | | | | Square III | | | | | |
|---|---|---|---|---|---|---|---|---|---|---|---|---|---|---|---|---|---|---|
| | Phase | | | | | | Phase | | | | | | Phase | | | | | |
| | 1 | 2 | 3 | 4 | 5 | 6 | 1 | 2 | 3 | 4 | 5 | 6 | 1 | 2 | 3 | 4 | 5 | 6 |
| 1 | $S_a$ | $S_b$ | $S_c$ | $S_d$ | $S_e$ | $S_f$ | $S_a$ | $S_e$ | $S_b$ | $S_c$ | $S_d$ | $S_f$ | $S_a$ | $S_d$ | $S_f$ | $S_e$ | $S_b$ | $S_c$ |
| 2 | $S_b$ | $S_d$ | $S_f$ | $S_a$ | $S_c$ | $S_e$ | $S_b$ | $S_f$ | $S_e$ | $S_d$ | $S_a$ | $S_c$ | $S_b$ | $S_f$ | $S_a$ | $S_c$ | $S_e$ | $S_d$ |
| 3 | $S_c$ | $S_f$ | $S_b$ | $S_e$ | $S_a$ | $S_d$ | $S_c$ | $S_a$ | $S_d$ | $S_e$ | $S_f$ | $S_b$ | $S_c$ | $S_b$ | $S_e$ | $S_f$ | $S_d$ | $S_a$ |
| 4 | $S_d$ | $S_a$ | $S_e$ | $S_b$ | $S_f$ | $S_c$ | $S_d$ | $S_b$ | $S_a$ | $S_f$ | $S_c$ | $S_e$ | $S_d$ | $S_e$ | $S_c$ | $S_a$ | $S_f$ | $S_b$ |
| 5 | $S_e$ | $S_c$ | $S_a$ | $S_f$ | $S_d$ | $S_b$ | $S_e$ | $S_c$ | $S_f$ | $S_a$ | $S_b$ | $S_d$ | $S_e$ | $S_a$ | $S_b$ | $S_d$ | $S_c$ | $S_f$ |
| 6 | $S_f$ | $S_e$ | $S_d$ | $S_c$ | $S_b$ | $S_a$ | $S_f$ | $S_d$ | $S_c$ | $S_b$ | $S_e$ | $S_a$ | $S_f$ | $S_c$ | $S_d$ | $S_b$ | $S_a$ | $S_e$ |

**Fig 3. Example set of three completely balanced Latin squares indicating the sequences in which standards are audited in Design B.**

possible (once or twice per phase in each stratum) in each of the 12 phase and stratum combinations. This balance can be achieved by using three completely balanced Latin squares, see e.g. Fig 3. Three completely balanced Latin squares are randomly selected, these are then randomised to the two strata, with at least one fully embedded in each of the two strata, e.g. Square I and three randomly selected rows of square II for stratum 1.

As in design A, in the first month of data collection at a facility, data are collected for all standards; however, in this design this month precedes the first month of data collection for the first standard to be audited. Similarly, in the last month of data collection at a facility data are collected for all standards audited, not just the standard for which data collection for the intervention state is due (see Figs 4 and 5).

## Other design variants

Two design variants have been described. For the standards developed within our Unit, the number of standards is more than the six considered here. Each variant can be used with more standards and the number of audit cycles planned for each facility could be changed, to better suit the context in which the trial is to be conducted.

Another variant using six standards per facility involves grouping the standards into six groups, according to the area of care; for each of the six areas of care each facility selects the standard in the group which they consider to be most in need of improvement.

In another variant audit cycles for two standards are conducted simultaneously in some of the facilities, e.g. those with higher staff numbers and thus able to handle the logistics of multiple audit processes. Note that for a single dimension the designs proposed are an expanded staircase design [10].

| Sequence | Stratum | Phase audited[a] | Month | | | | | | | | | | | | | | | |
|---|---|---|---|---|---|---|---|---|---|---|---|---|---|---|---|---|---|---|
| | | | 0 | 1 | 2 | 3 | 4 | 5 | 6 | 7 | 8 | 9 | 10 | 11 | 12 | 13 | 14 |
| 1 | I | 1 | 0 | 0 | T | 1 | | | | | | | | | | 1 | |
| 2 | II | 1 | 0 | | 0 | T | 1 | | | | | | | | | | 1 |
| 3 | I | 2 | 0 | | | 0 | T | 1 | | | | | | | | 1 | |
| 4 | II | 2 | 0 | | | | 0 | T | 1 | | | | | | | | 1 |
| 5 | I | 3 | 0 | | | | | 0 | T | 1 | | | | | | 1 | |
| 6 | II | 3 | 0 | | | | | | 0 | T | 1 | | | | | | 1 |
| 7 | I | 4 | 0 | | | | | | | 0 | T | 1 | | | | 1 | |
| 8 | II | 4 | 0 | | | | | | | | 0 | T | 1 | | | | 1 |
| 9 | I | 5 | 0 | | | | | | | | | 0 | T | 1 | | 1 | |
| 10 | II | 5 | 0 | | | | | | | | | | 0 | T | 1 | | 1 |
| 11 | I | 6 | 0 | | | | | | | | | | | 0 | T | 1 | |
| 12 | II | 6 | 0 | | | | | | | | | | | | 0 | T | 1 |

**Fig 4. Schematic representation for design B of data collection and action for standard $S_a$.** 0 denotes assessment of compliance with the standard under the current standard of care. T denotes a transition month in which action is taken after review of data collected previously. 1 denotes assessment of compliance after taking action to improve the quality of care delivery for the standard. Within each sequence the audit cycle periods are indicated by boldly bordered sets of three-month periods.

## Design: Sample size

Quantities which need to be specified when designing a M-DISW-CRT trial for Standards-based audits include: the number of cluster (facilities) to participate; the number of standards / audit cycles per facility; and the number of clients whose care is to be assessed per assessment month per facility.

For any given context in which this trial design is to be used the number of clusters, the time period available in which to conduct the trial and the number of standards may be pre-determined. The first number to determine is the number of facilities to participate. In order

| Representation | Standard | Month | | | | | | | | | | | | | |
|---|---|---|---|---|---|---|---|---|---|---|---|---|---|---|---|
| | | 0 | 1 | 2 | 3 | 4 | 5 | 6 | 7 | 8 | 9 | 10 | 11 | 12 | 13 |
| By standard | $S_a$ | X(a) | X(a) | T(a) | X(a) | | | | | | | | | | X(a) |
| | $S_b$ | X(b) | | | X(b) | T(b) | X(b) | | | | | | | | X(b) |
| | $S_c$ | X(c) | | | | | X(c) | T(c) | X(c) | | | | | | X(c) |
| | $S_d$ | X(d) | | | | | | | X(d) | T(d) | X(d) | | | | X(d) |
| | $S_e$ | X(e) | | | | | | | | | X(e) | T(e) | X(e) | | X(e) |
| | $S_f$ | X(f) | | | | | | | | | | | X(f) | T(f) | X(f) |
| As a facility | All | X(a,b,c,d,e,f) | X(a) | T(a) | X(a,b) | T(b) | X(b,c) | T(c) | X(c,d) | T(d) | X(d,e) | T(e) | X(e,f) | T(f) | X(a,b,c,d,e,f) |

**Fig 5. Schedule of assessments for the facility in row 1 of square I in design B.** X(a) indicates that data for $S_a$ was to be collected in that month. T(a) indicates that data for $S_a$ was to be reviewed and action taken. X(b), X(c), X(d), X(e), X(f) and T(b), T(c), T(d), T(e), T(f) are similarly defined. X(a,b,c,d,e,f) indicates that data for $S_a$, $S_b$, $S_c$, $S_d$, $S_e$ and $S_f$ was to be collected in that month.

to avoid the multiple risks associated with using few clusters [14,15] we specified that the number of facilities recruited to participate should not be fewer than 18.

In clinical practice the size of sample to use in each data collection phase, when performing a S-BA, depends on the rate at which clients become available, the desired duration of data collection phases and the degree of improvement desired to be achieved. Primarily for practical reasons, for the purposes of our trials a fixed sample size of 25 clients for each step (of fixed one month duration) for a standard was used. Though this is smaller than sample sizes typically used in clinical practice (a systematic review of clinical audits to assess quality of obstetrical care in low- and middle-income countries reported 10 studies with a median sample size per collection phase of 403 [4]) it was considered sufficient in the context of the trial. This resulted in an inclusion criterion that participating facilities should expect to have at least 25 clients available in each month.

The R shiny app developed by Karla Hemming and Jessica Kasza (available at https://clusterrcts.shinyapps.io/rshinyapp/) to perform sample size and power calculations for a range of trial designs, including incomplete stepped wedge trials, [16] was used for sample size calculations. Although the R shiny app is designed for use with a simpler design of study the app was used for this trial design, by treating the distinct standards within a common facility as being from distinct facilities. This approach ignores the fixed effect of standard which is assumed in model fitting, but is more accurate than the alternative of inflating the number of replications by the number of audit cycles to be performed, since it accounts partially for the differing underlying levels of compliance between standards.

For Designs A and B the design matrix needed for sample size calculations can be obtained from Figs 1 and 4 respectively. To calculate the power / difference detectable using this design the number of clusters specified per sequence in the design matrix then needs to be inflated by a factor of the number of audit cycles per facility. Thus, for example in using design B with 12 sequences and 6 audit cycles planned in each of the 18 facilities the number of clusters per sequence, specified when using the app, was 9 (= 6x18/12). A cross-sectional sampling structure is required, since clients are not followed longitudinally for data collection; for the

correlation structure the discrete-time decay option allows correlation in responses to attenuate over time. Since this design of trial is novel no estimates for the within period intra-cluster correlation coefficient and the cluster auto correlation coefficient are available. Therefore, a range of plausible values should be used to ensure that the design used can be expected to be sufficiently robust.

The primary outcome (compliant or not, irrespective of standard) is binary. When commencing the trial the compliance level with any given standard will usually be unknown, but could be 50%. To ensure that the difference detectable is not under-estimated the proportion under control should therefore be set at 50%, unless there are clear reasons to expect this to be implausible, e.g. available data suggest a different compliance level.

## Trial conduct: Participant recruitment and blinding

As far as possible recruitment of clients to participate in the trial should follow the standards used in performing a clinical audit. Thus, chronological recruitment starting on the first day of the relevant study month, until the required sample size is obtained would usually be appropriate. Clients would only be asked to consent for data to be collected for a standard which requires that data be collected directly from clients, e.g. by interview regarding respectfulness of care received. For each standard there may be distinct inclusion and exclusion criteria, e.g. for the standard "*All women who require caesarean section are given prophylactic antibiotics*" only women on whom a caesarean section was performed would be recruited. When HCPs perform a clinical audit it is not possible for them to be blinded to whether the intervention has been delivered or not. The same applies for the conduct of a M-DISW-CRT trial for S-BAs.

Occasionally a client may provide data within a month for two (or more) standards, or in some cases in different months. Compliance with standards is a function of the HCPs involved in delivery of care rather than the recipient of the care and thus linking data by client would be of no statistical value (and would increase the complexities of data collection). Inclusion of a term in analysis to account for the HCPs involved in care delivery is not proposed since such data would not normally be considered in the conduct of clinical audit. Additionally, any attempt to collect such data would be detrimental to the quality of data if doing so triggers connotations of witch-hunting.

## Analysis

### Statistical methods to assess intervention effect

If the trial is completed without deviation there should be 25 individual records, each classified as compliant or otherwise with the standard, for each combination of facility, month and standard in which data are collected. In design A, for each facility, there are three months for each standard in which data are obtained for 25 clients, each having control (0) or intervention status (1) as in Fig 1. For design B, there are four months, except when the standard is audited in the final phase of the trial, when there will only be three months (sequences 11 and 12), as in Fig 4.

An overall estimate of the benefit of performing Standards-based audit can be derived using a generalised linear mixed model framework (also known as a multilevel modelling approach) to examine the evidence of intervention effects using data aggregated across all standards. Generalised linear mixed models are appropriate in the context of stepped-wedge trial designs as they allow for clustering and address confounding of intervention effects and time [17]. For the binary primary outcome measure data for each client assessed should be aggregated for all standards and a logit link function used. The effect of the intervention can then be

estimated, using generalised linear mixed models, and reported as an odds ratio, with a 95% confidence interval [15].

In addition to the binary intervention effect the primary analysis should also include several other terms. As established by Hussey and Hughes [18] step (month of the trial in which the care assessed was provided) should be included as a fixed effect, to account for underlying secular trend, and clusters (facilities) should be random effects. Standards should also be included as fixed effects, to account for variation in baseline levels of compliance between standards. Hemming, Taljaard and Forbes [17] describe several model extensions to account for heterogeneity in secular trends. Their model extension B includes random effects for study month by facility interaction to account for variation between facilities in secular trends. The month-by-facility interaction is expected to be applicable to this design and should therefore be included in the primary analysis (reference model).

## Sensitivity analysis terms

There are several other interaction terms that it is useful to consider including in sensitivity analyses. Interactions involving the intervention effect are considered first, and then other interactions.

Interaction between intervention and study month is an extension (E) considered by Hemming, Taljaard and Forbes [17]. However, even in the context of the more complete study they used, inclusion of this interaction effect was uninformative. In our incomplete design with sparser data we do not expect it to be of interest. There is an inherent association between intervention and time in all stepped wedge study designs. We therefore do not recommend inclusion of this term in the primary analysis.

Inclusion of interaction between intervention and standard would posit that the benefit of conducting Standards-based audits is specific to the standard involved. Consideration of this hypothesis may be useful to explore the assumption of the study design to the contrary.

Since the focus is on the benefits of introducing the S-BA process at any facility the facility term is random and thus exploration of a possible interaction between intervention and facility is not of value.

Intrinsically it is anticipated that there is some variation between the facilities in the standards which are well complied with. This suggests an interaction between facility and standard, which could be considered as an additional random effect, to account for any such interaction.

The sparseness of the design in terms of months in which data are collected for each standard is a limitation which impedes the capacity of this study design to estimate multiple interaction terms. Thus, for example in the use of design A with a larger pool of standards model fitting did not converge when the random effect for interaction between facility and standard was added to the model. It was necessary to drop the fixed effect for study month from the model to include the interaction terms for facility by study month and facility by standard for model fitting to converge. (The random effect for facility was also dropped.) Finally, interaction between study month and standard would account for variation between standards in secular trends.

## Secondary outcome measures

Although the intention in designing the trial is that it is balanced for each standard assessed, the data points for each individual standard are relatively sparse, with a single cluster dropping out having the potential to result in a step in which only one facility contributes data. Thus, they are prone to design imbalances which may result in spurious statistically significant findings. For this reason we recommend that analysis of individual standards is not routinely planned; if it is planned it should only be as a secondary analysis and interpreted with caution.

## Discussion

### What is estimated

The multi-dimensional incomplete stepped-wedge design presented is a novel randomised trial design with potential for use in deriving an unbiased estimate of the benefit of introducing the S-BA process into clinical practice in health facilities. The measure used is specific to implementation of the process for individual (though unspecified) standards, and not to be interpreted as a measure of compliance with all standards from introducing the S-BA process.

A defining characteristic of this trial design is that the intervention for which the impact is being assessed is not a unique pre-defined set of actions to which participating facilities are randomised. The objective is to estimate the impact of introducing S-BA as a process, on compliance with any standard selected for audit. As is the case in any randomised trial some variation between participants is expected; the focus in this design is on estimation of the effect, regardless of the standard(s) selected for audit. In this trial design, a standard would not be selected unless compliance is perceived to be less than satisfactory and there is potential for a detectable improvement.

There is no intrinsic interest in the trial design in the effect of individual standards, so they might be considered as random effects. However, their number is limited, and each standard is selected purposively for use by the facilities at which it is used, so the effects of standards are treated as fixed.

A secondary objective in the conduct of this design of trial is that HCPs gain knowledge and skills to adopt the practice of performing S-BA in their clinical practise, beyond the duration of the trial. The design of the trial sought to promote this, but some features may limit the manner in which it is adopted and therefore it would be important to acquaint them with these issues during the closing phase of the trial:

a. The number of cases reviewed in each step was set at 25. If used in a single facility this sample size would require an improvement of more than 40% in compliance for there to be 80% power to detect change. For some standards with compliance below 60% this may be a reasonable target, but for others it may be overly ambitious.

b. To enable a stepped wedge design to be implemented the time period in which an audit cycle is performed is standardised. However, for some standards in facilities handling smaller numbers of clients a longer time period may be required to accumulate the required volume of data.

c. In some contexts the audit cycle process will need to be iterated (as is very common in the clinical implementation of the clinical audit process).

### Duration of effect

By definition a stepped-wedge design assumes that once the intervention has been delivered the cluster is in the "intervention state". The effect estimated is that of the odds of compliance with a standard resulting from applying the process to a standard. Since the standards selected are expected to be those in most need of action and thus with greatest scope for improvement it may also be true that the benefit is greater and thus that the benefit of adoption of the process in clinical practice is over-estimated. For any given standard there may be a drift back over time to a compliance level similar to that in the Control phase, due for example to staff turnover. Therefore, as in any clinical trial the estimated benefit should only be extrapolated beyond the study duration with caution.

### Inclusion of baseline and endline assessments

In both designs data are collected for each standard according to the S-BA design and additionally, during the first and last steps for the facility. These additional rounds of data collection were included to estimate any underlying secular trends more accurately and thereby improve the precision of estimates of effect. An additional benefit of endline assessments is that they provide opportunity to examine evidence of amplification or attenuation of effect over time as a main effect. In design A the first step is the first step of an audit cycle, whereas in design B it is not. This provides an opportunity to review the 'underlying / pre-study' levels of compliance with standards selected for audit. This has both a benefit and a challenge. There is the benefit of providing opportunity for the set of standards selected for use at a facility to be revised, as prescribed in the trial protocol, in the event that the compliance is already satisfactory (analysis plans need to be specified to address such data driven changes). The challenge arises when the compliance is deficient and, having recognised the problem, there is capacity for the HCPs to improve compliance without waiting to complete the audit cycle when scheduled. This would over-estimate compliance in the control state and thus under-estimate the impact of the process.

### Stratification

For design A the lack of overlap of audit cycles within facilities necessitated stratification to eliminate confounding between step and intervention. When the audit cycles overlap such confounding is not present and stratification is not necessary, although it was used in design B. Without stratification all facilities have the same pattern over all steps of collecting data and reviewing findings and implementing changes to address deficiencies.

### Sample size

We have conducted trials using design variants A and B. For Design A in Malawi 44 facilities were available and used, whereas for the trial using design B in Togo we were limited to 18 facilities. Both trials detected an improvement in compliance with standards with the ORs (95% CI) for compliance associated with application of the S-BA process to Standards estimated to be 2.80 (1.65,4.76) in the trial using design A and 3.49 (2.13,5.74) in the trial using design B.

### Contribution to Clinical audit toolkit for developing countries

Several groups have explored various approaches to the promotion of clinical audit in the African continent. Wekesah and colleagues [5] highlight the important role that improvements in the health care system can contribute to improve maternal health care. Weeks and colleagues [19] note the potential of improvements at low cost that adoption of S-BA can provide, but also highlight the crucial role of follow up to prevent a waning of enthusiasm with time. Grimshaw and colleagues [20] emphasize the need to understand how to optimise audit and feedback to maximise its effects. While the stepped-wedge design we propose provides one approach to estimate the benefit of completing standards-based audits in a way which is expected to both foster the adoption of the process of conducting standards-based audits among HCPs, as well as providing a methodology for estimation of the benefits of audit more research is needed to ensure that the introduction and development of a culture of using S-BA in clinical practise is optimised.

## Limitations

To examine evidence of interaction between standard and facility other terms had to be dropped from the model. This is because of the sparseness of the study design. It is thus not possible to examine whether, after accounting for the interaction of standard and study month, there is also evidence of an interaction between facility and standard. The sparseness of the design also limits the opportunities to examine the data for evidence of bias arising from order effects associated with particular standards.

This design focuses only on the standards of interest. It is, therefore, possible that an improvement in compliance with standards selected has a detrimental impact on the delivery of other standards, or on some other aspect of care not captured in the standards. The design of this study does not enable such impacts to be estimated. Similarly, it is not proposed that this trial design be used to examine interactions involving fixed effects. If such interactions are to be accounted for a more complex study design is required.

When using this design of trial to assess the impact of introducing S-BA there are two potential limitations which need to be considered. There are often external factors, such as non-availability of test kits / drugs / equipment required for a standard to be delivered, over which HCPs at participating facilities have no potential / capacity to improve. If the current level of compliance with the standard is 'too high' there may not be sufficient scope for an improvement to be detected.

The proposed analysis assumes that there is no attenuation / amplification of benefit for the intervention over time. This may be an unrealistic assumption. The potential to include a covariate to measure a time lag following the intervention to model this may be considered.

## Conclusion

The multi-dimensional stepped-wedge design described is a novel trial design. It can be used to estimate the impact of introducing the S-BA process on compliance with standards when an audit cycle is conducted on a standard that has been identified as suitable for audit because of perceived deficiencies in compliance and the potential for improvement.

Approaches to sample size estimation and statistical analysis using available software have been described. The design may have applications in other contexts in which multiple processes are to be studied. The sample size used should be at least 18 clusters. This is in line with recommendations in more standard cluster randomised trials in which any incompleteness is generally due to non-assessment during transition periods [15]. The data collection in the design presented is sparser per individual dimension than in a complete design. But when aggregated across standards the design involves data collection in the majority of steps. However further research to assess the robustness of the design in terms of the numbers of clusters (facilities) participating and the numbers of standards audited would be useful.

## Acknowledgments

Nynke van den Broek posed the initial question which led to the development of this trial design. Without her challenge and encouragement and the opportunity to deploy it in projects undertaken within projects undertaken in the Department of International Health it would not have been conceived.

Barbara Madaj commented on a preliminary draft.

Richard Hooper and Brian Faragher made valuable comments on a full draft prior to submission.

## Author Contributions

**Conceptualization:** Sarah Ann White.

**Formal analysis:** Sarah Ann White.

**Investigation:** Sarah Ann White.

**Methodology:** Sarah Ann White.

**Writing – original draft:** Sarah Ann White, Florence Mgawadere.

**Writing – review & editing:** Sarah Ann White, Florence Mgawadere.

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
