## [Decision Letter · Decision Letter 0]

23 Aug 2023

PONE-D-23-11849A multi-dimensional incomplete stepped-wedge trial design to estimate the impact of standards-based auditPLOS ONE

Dear Dr. Sarah Ann White

Thank you for submitting your manuscript to PLOS ONE. After careful consideration, we feel that it has merit but does not fully meet PLOS ONE’s publication criteria as it currently stands. Therefore, we invite you to submit a revised version of the manuscript that addresses the points raised during the review process.

Please submit your revised manuscript within two weeks. If you will need more time than this to complete your revisions, please reply to this message or contact the journal office at plosone@plos.org. Please include the following items when submitting your revised manuscript:A rebuttal letter that responds to each point raised by the academic editor and reviewer(s). You should upload this letter as a separate file labeled 'Response to Reviewers'.A marked-up copy of your manuscript that highlights changes made to the original version. You should upload this as a separate file labeled 'Revised Manuscript with Track Changes'.An unmarked version of your revised paper without tracked changes. You should upload this as a separate file labeled 'Manuscript'.We look forward to receiving your revised manuscript.

Kind regards,

Mergan Naidoo, PhD

Academic Editor

PLOS ONE

Journal Requirements:

Reviewers' comments:

Reviewer's Responses to Questions

**Comments to the Author**

1. Is the manuscript technically sound, and do the data support the conclusions?

Reviewer #1: Yes

Reviewer #2: Yes

2. Has the statistical analysis been performed appropriately and rigorously? 

Reviewer #1: Yes

Reviewer #2: N/A

3. Have the authors made all data underlying the findings in their manuscript fully available?

Reviewer #1: Yes

Reviewer #2: Yes

4. Is the manuscript presented in an intelligible fashion and written in standard English?

Reviewer #1: Yes

Reviewer #2: Yes

5. Review Comments to the Author

Reviewer #1: Well written and fairly explained. Just I would like to know more about considerations after those limitation points. For example, you have mentioned in line 502 (the covariate maybe considered) and that (in my opinion) will let the reader be thrilled and imagining what will these improvements do next?

Overall, Nice article

Reviewer #2: This study is highly intriguing as it employs a complex research design to investigate the impact of standards-based audits. The design incorporates multiple dimensions and utilizes a stepped-wedge trial approach. In this approach, the application of the intervention (standards-based audits) varies across different clusters, rather than being uniform. The primary objective is to estimate the effects of these audits on specific outcomes.

The study introduces an innovative trial design that avoids the common pitfalls associated with the use of before-after designs in clinical audits, which often lead to biased impact estimations. However, there are certain aspects that, in my opinion, the authors should carefully contemplate before proceeding with the publication of their work in your journal.

Neither in the introduction nor in the discussion sections did the authors clearly explain why they believe the chosen design would be more advantageous when compared to existing approaches (such as before and after designs). I would appreciate it if the authors could offer readers a comprehensive justification for the trial design and elaborate on its distinctions from the before-after design used in clinical audits. What limitations of the before-after design have influenced their preference for the trial design?

In the body text, I recommend placing tables immediately after their corresponding citations. This approach not only adheres to the journal's guidelines but also enhances the reader's comprehension flow.

I am of the opinion that there are certain writing issues that should be addressed on a line-by-line basis. For instance, in line number 115 of the manuscript, the in-text citation format appears inconsistent with other citations, and information source for Table 1 is labeled as "source:2".

Overall, this is a commendable piece of work with significant novelty, and it deserves support. If published, it has the potential to benefit numerous researchers and contribute to the ongoing improvement of our audit practices, particularly in developing countries.

6. PLOS authors have the option to publish the peer review history of their article (what does this mean?). If published, this will include your full peer review and any attached files.

Reviewer #1: No

Reviewer #2: No

---

## [Author Response · Author response to Decision Letter 0]

26 Sep 2023

Reviewer’s comment 

Well written and fairly explained. Just I would like to know more about considerations after those limitation points. For example, you have mentioned in line 502 (the covariate maybe considered) and that (in my opinion) will let the reader be thrilled and imagining what will these improvements do next? 

Response:

Thank you for your interest in our manuscript and your desire to know more. The point noted is a comment regarding a possible approach to the modelling of a particular circumstance. Our aim in the manuscript is to present the design and the potential value of using the design. 

In the example noted we suggest a possible approach to handling a particular circumstance; the potential value of the approach would need to be explored in the context of a particular study design. We have not amended the text in response.

Reviewer’s comment:

Neither in the introduction nor in the discussion sections did the authors clearly explain why they believe the chosen design would be more advantageous when compared to existing approaches (such as before and after designs). I would appreciate it if the authors could offer readers a comprehensive justification for the trial design and elaborate on its distinctions from the before-after design used in clinical audits.

Response:

Thank you for highlighting the need to provide justification for the design and distinguish it from the before-after design. Text has been added as follows:

Lines 105-109 indicate the ways in which the SW-CRT design is related to the before-after design but also provides a mechanism for addressing any bias in estimation due to a secular trend, which the before-after design cannot do.

Lines 152-3 have been added to clarify an aspect of the extension that should be mentioned.

Reviewers comment:

What limitations of the before-after design have influenced their preference for the trial design? 

Response:

Sentences have been added in lines 55-57 to indicate the limitations and line 87 has been modified.

Reviewer’s comment:

In the body text, I recommend placing tables immediately after their corresponding citations. This approach not only adheres to the journal's guidelines but also enhances the reader's comprehension flow. 

Response:

The guidelines indicate that each table should be placed immediately after the paragraph in which the first citation of a table occurs. I have verified that this is how they are positioned (and inserted some page breaks to avoid tables and Figures being split across pages).

Reviewer’s comment:

I am of the opinion that there are certain writing issues that should be addressed on a line-by-line basis. For instance, in line number 115 of the manuscript, the in-text citation format appears inconsistent with other citations, and information source for Table 1 is labeled as "source:2". 

Response:

Thank you for identifying these issues which have now been addressed (line 115 is now line 121).

Table 1 has been revised to acknowledge the content from the reference cited more accurately in the format used in Table 2 (as no comment was raised for Table 2 I presume it is correct; I can’t find guidance on this in the webpages).

Reviewer’s comment:

Overall, this is a commendable piece of work with significant novelty, and it deserves support. If published, it has the potential to benefit numerous researchers and contribute to the ongoing improvement of our audit practices, particularly in developing countries. 

Response:

Thank you for your endorsement of the value of this work.

Editors Comment:

Response:

These have now been checked, and Figures no longer contain legends etc

Editors Comment:

Please review your reference list to ensure that it is complete and correct. If you have cited papers that have been retracted, please include the rationale for doing so in the manuscript text, or remove these references and replace them with relevant current references. 

Response:

All references were checked on 4th Sept 2023. Some details have been revised to ensure compliance with standards for references.

The reference we numbered 1 is no longer available directly via the WHO webpages. We have therefore referenced an alternative (previously numbered 6). This change has necessitated some revision of reference numbers (one of which was identified to have been incorrect).

---

## [Editor Report · Decision Letter 1]

31 Oct 2023

A multi-dimensional incomplete stepped-wedge trial design to estimate the impact of standards-based audit

PONE-D-23-11849R1

Dear Dr. Sarah Ann White

We’re pleased to inform you that your manuscript has been judged scientifically suitable for publication and will be formally accepted for publication once it meets all outstanding technical requirements.

Kind regards,

Mergan Naidoo, PhD

Academic Editor

PLOS ONE
---

## [Editor Report · Acceptance letter]

20 Nov 2023

PONE-D-23-11849R1 

A multi-dimensional incomplete stepped-wedge trial design to estimate the impact of standards-based audit 

Dear Dr. White:

I'm pleased to inform you that your manuscript has been deemed suitable for publication in PLOS ONE. Congratulations! Your manuscript is now with our production department. 

Kind regards, 

on behalf of

Professor Mergan Naidoo 

Academic Editor

PLOS ONE